# Assessing Stroke-Related Sarcopenia in Chronic Stroke: Identification of Clinical Assessment Tools—A Pilot Study

**DOI:** 10.3390/biomedicines11102601

**Published:** 2023-09-22

**Authors:** Anna Arnal-Gómez, Sara Cortés-Amador, Maria-Arantzazu Ruescas-Nicolau, Juan J. Carrasco, Sofía Pérez-Alenda, Ana Santamaría-Balfagón, M. Luz Sánchez-Sánchez

**Affiliations:** 1Physiotherapy in Motion, Multispeciality Research Group (PTinMOTION), Department of Physiotherapy, Faculty of Physiotherapy, University of Valencia, Gascó Oliag n 5, 46010 Valencia, Spain; anna.arnal@uv.es (A.A.-G.); arancha.ruescas@uv.es (M.-A.R.-N.); juan.j.carrasco@uv.es (J.J.C.); sofia.perez-alenda@uv.es (S.P.-A.); m.luz.sanchez@uv.es (M.L.S.-S.); 2Faculty of Physiotherapy, University of Valencia, Gascó Oliag n 5, 46010 Valencia, Spain; asanbal@alumni.uv.es

**Keywords:** stroke, sarcopenia, symptom assessment, clinical tools, neurologic examination, muscle strength, walking speed, motor skills disorders, physical disability, motor impairment

## Abstract

Stroke-related sarcopenia has recently been defined as the muscle atrophy consequent to stroke and assessing it following the guidelines with simple clinical tools is crucial in chronic stroke survivors. The aim of this study was to determine the characteristics of patient-friendly instruments sarcopenia in a chronic stroke sample (SG) compared to non-stroke counterparts (CG). Each participant underwent a single assessment which consisted of: SARC-F questionnaire, assessment of muscle strength (hand grip and five-times sit-to-stand test, 5STS), the calf circumference (CC) of both legs, the short physical performance battery (SPPB), and the 10 m walk test. A total of 68 participants were included (SG, n = 34 and CG, n = 34). All variables showed statistical differences (*p* < 0.05) between the SG and the CG, except handgrip although it showed lower values for SG. The values of the 5STS (16.26 s) and the SPPB (7 points) were below to the cutoff values for the SG. The five-times sit-to-stand test, SPPB, and gait speed can lead clinicians to detect stroke-related sarcopenia. Maximum handgrip shows a trend of low values for men and women in the SG, however, CC did not detect sarcopenia in our sample.

## 1. Introduction

Stroke can be triggered by an ischemic infarct in one or several brain areas that ends with cell death, or by the rupture of a blood vessel in the cerebral vascular territory [1]. Although the mortality rate from stroke has decreased over time, the increase in life expectancy and risk factors contribute to the fact that the prevalence of stroke has remained high [2], and it is still the second leading cause of death [3]. For those who survive, recovery is heterogeneous in terms of outcomes since it is estimated that 50% of stroke survivors worldwide will suffer chronic disability [4]. The most frequent symptoms and signs in people who have suffered a stroke are motor skills disorders (predominant unilateral motor weakness, muscle tone alterations, such as spasticity), sensory deficit, facial asymmetry (weakness and/or paralysis), consciousness alterations, speech (aphasia and/or dysarthria) and swallowing (dysphagia) alterations, visual disturbances (diplopia, blurred vision), headaches, and balance and gait disorder. The most common impairment is unilateral paresis or hemiparesis on the contralateral side to where the brain injury occurred [5]. Therefore, the long-term disability caused by stroke is largely due to an impairment of motor function [6], which probably has a multi-factorial cause related to decreased descending drive and disuse (due to reduced physical activity and compensatory motor patterns) and spasticity, that lead to muscle atrophy and weakness [7].

In the aging process, muscle tissue is gradually lost, resulting in a decrease in strength and mass, which is a syndrome described as sarcopenia [8]. The European Working Group on Sarcopenia in Older People 2 (EWGSOP2) defines sarcopenia as geriatric disease characterized by a progressive and widespread loss of skeletal muscle mass and strength, with an increased risk of adverse events related to mobility, such as physical disability, falls, hospitalization, and death. Some markers for sarcopenia are low grip strength, low muscle density, and slow walking speed [9], with low muscle strength outweighing the role of low muscle mass as the main determinant [8]. Sarcopenia caused by aging is called primary sarcopenia, it is a manifestation of the aging process of the body without another evident specific cause, and it is common in the older adult population. The term secondary sarcopenia is applied when factors other than aging are evident, such as inactivity, disease, and malnutrition, although often there is not a clear clinical difference between the two types [10]. Sarcopenia related to disease accelerates the progression of muscle atrophy and becomes part of the pathology process.

Stroke often produces brain injury resulting in the loss of brain functions, particularly motor function [11]. However, the loss of systemic muscle strength and mass, and decreased function after stroke, cannot be explained by brain injury alone [12]. Therefore, the muscle atrophy consequent to stroke has recently been defined as “stroke-related sarcopenia” and belongs to the secondary sarcopenia group [6]. Stroke-related sarcopenia, as it has been described, reveals features that distinguish it from age-related sarcopenia: a quick decline of muscle mass following the stroke incident, plus structural muscle alterations (muscle fibers shift from slow-twitch fibers to fast-twitch fibers). There is a descending neural output, caused by stroke, which leads to functional and structural changes in skeletal muscle [6]. This descending drive refers to the neural signals sent from the brain to the spinal cord and peripheral muscles to initiate and control movement. Stroke-induced brain damage can disrupt these neural pathways, leading to impaired motor control and reduced muscle activation [13]. Understanding these neural and metabolic mechanisms can aid in optimizing motor recovery and minimizing the adverse effects of sarcopenia in chronic stroke survivors [14,15,16]. Moreover, disuse and compensatory motor patterns may arise due to decreased physical activity following stroke, contributing to muscle atrophy and weakness Common symptoms in people with stroke are fatigability, weakness, hypotrophy, and altered motor control, resulting from the combination of denervation, disuse, remodeling, and spasticity [17]. Continuous spastic muscle activity can cause muscle fatigue and atrophy over time, further exacerbating the muscle weakness seen in stroke-related sarcopenia [14]. Overall, this will determine the bilateral differences in physical and functional performance depending on the brain lesion [18].

Recent clinical studies have shown that the prevalence of sarcopenia after stroke ranges from 14% to 18%, and it is expected to increase in the next 20 years [19]. Moreover, in the chronic stage after stroke, the decrease in motor neurons continues further, leading patients with chronic stroke to higher rates of sarcopenia compared with healthy individuals [15,16]. However, most of the stroke-related sarcopenia studies have focused on the acute stage and a recent study has highlighted there is a lack of clinical literature on post-stroke sarcopenia in the chronic stage [20,21] and, therefore, the understanding of its epidemiology, screening, evaluation, and treatment is still limited [12].

Considering that both the neural deterioration and the systemic loss of muscle strength and mass in stroke-related sarcopenia may decrease the ability of patients to perform activities of daily living (ADL) [22], it is of paramount importance to screen and measure these variables in the clinical setting in order to establish the correct prevention and treatment. The evaluation of sarcopenia is based preferably on the guidelines from the European Working Group on Sarcopenia in Older People (EWGSOP2) [8] or from the Asian Working Group for Sarcopenia [23]. The evaluation tools stated in these guidelines focus on the assessment of muscle strength, detection of muscle quantity and quality, and identification of physical performance. Taking into account that some of the techniques to measure these variables can be time- and resource-demanding (e.g., bioimpedance analysis, dual-energy X-ray absorptiometry, computerized tomography or magnetic resonance imaging), the routine assessment for sarcopenia of chronic stroke survivors with these full-diagnostic measures may not be feasible in everyday clinical settings, which, in turn, makes it difficult to detect cases of sarcopenia [24]. Therefore, measurement of sarcopenia variables following the guidelines but considering patient-friendly instruments in the clinical context are crucial to detect stroke-related sarcopenia, thus, decreasing adverse outcomes.

Previous literature has studied some measurements in stroke survivors, such as hand grip [25] strength or calf circumference [26], in an isolated manner. However, to the best of our knowledge, no previous study has analyzed these sarcopenia variables on the whole. Moreover, muscle strength declines around the age of 40 years, plus, the physiological decline may vary between sexes [27], therefore, matching stroke patients with age and sex counterparts can reveal clinical aspects still to be elucidated. Hence, we tested the hypothesis that clinically patient-friendly instruments for assessing sarcopenia will have different ability for screening sarcopenia compared to age and sex counterparts. Thus, the aim of this study was: (1) to analyze characteristics of the measurement of sarcopenia variables with clinically tools in a chronic stroke sample and compare them to non-stroke counterparts; and (2) to compare the characteristics of these measurements on stroke and non-stroke participants depending on their age group (40–65 years old, and >65 years old).

## 2. Materials and Methods

### 2.1. Study Design and Participants

Between September 2021 and December 2022, a cross-sectional study was conducted in the Valencia region (Spain). The study focused on adults aged 40 years and older who had chronic stroke (onset ≥ 6 months), as well as a control group of non-stroke individuals. In order to eliminate potential confounders of sarcopenia, stroke survivors were matched with non-stroke control counterparts on the basis of sex and age ±2 years.

Participants with chronic stroke (referred to as the stroke group, SG) were recruited from brain injury associations, everyone, following the recommendations of the Valencian community, received rehabilitation treatment in both the acute and chronic phases [28], while non-stroke participants (control group, CG) were recruited through advertisements at the faculty (administrative workers, teachers, and their families) and from associations of older adults. To be included in the study, participants had to be able to walk independently indoors, with or without mobility assistance, and without supervision from another person, and had to live in the community. Participants of both groups were excluded if they had an acute illness, experienced significant pain (visual analogue scale > 5), or had cardiac, locomotor, neurological, or cognitive impairments that would hinder their ability to complete the assessment tests, questionnaires, and provide informed consent.

Before being included in the study, participants were given detailed information about the study’s purpose and experimental procedures, and they provided written informed consent. The study received approval from the Ethics Committee for Human Research at the University of Valencia (ID no. 1563377228465) and followed the principles outlined in the Declaration of Helsinki [29]. The design, conduct, and reporting of the study were guided by the Strengthening the Reporting of Observational Studies in Epidemiology (STROBE) guidelines [30].

### 2.2. Procedures

Each participant underwent a single evaluation session for a period of one hour and a half. The session began by gathering demographic and clinical information through medical records and clinical interview. Measurements of weight and height were then taken. Subsequently, cognitive function was assessed by using the Montreal Cognitive Assessment (MOCA) [31] and the level of functional independence was measured with modified Rankin scale [32]. To identify probable sarcopenia, the strength, assistance in walking, rise from a chair, climb stairs, falls history questionnaire (SARC-F) [33] was used. The evaluation continued with the assessment of muscle strength (hand grip, bilaterally) [34] and five-times sit-to-stand test [35], and then measuring the calf circumference of both legs [36]. This was followed by the evaluation of physical function using the short physical performance battery (SPPB) [37] and the 10 m walk test (10 MWT) [38].

### 2.3. Outcome Measures

#### 2.3.1. Sarcopenia Screening: Strength, Assistance in Walking, Rise from a Chair, Climb Stairs, Falls History Questionnaire (SARC-F)

The SARC-F [33] is a simple questionnaire, useful to predict physical limitation, and which consists of five items that assess strength, assistance with walking, rising from a chair, climbing stairs, and falls. It is designed to capture key aspects and outcomes related to sarcopenia. The scale score spans 0 to 10, with each component being allocated 0–2 points. A score of 0 indicates the best condition, while a score of 10 indicates the worst. Scores between 0 and 3 indicate a healthy status, while scores of 4 or more signify the presence of sarcopenia [39]. SARC-F scale showed an adequate internal consistency (α = 0.64) and reliability test retest (ICC = 0.80) [33].

#### 2.3.2. Muscle Strength: Hand Grip and Five-Times Sit-to-Stand Test

The maximum hand grip [34] was measured by using a hydraulic hand dynamometer Jamar Plus+ (Patterson Medical, Sammons Preston, Bolingbrook, IL, USA, EE. UU). The standardized procedure for positioning of the instrument was performed following the American Society of Hand Therapists protocol [40]. The patient was seated with his/her shoulder adducted and neutrally rotated, elbow flexed at 90°, and the forearm and wrist in neutral position [40]. The maximal grip strength was measured alternatively three times in each arm, with the highest measurement being recorded. For the SG, hand grip was measured on the paretic and non-paretic side. Low muscle strength was referred to <27 and <16 kg for men and women, respectively [41]. Additionally, the hand grip strength difference between limbs was calculated. Hand grip has shown an excellent intrarater reliability in both people with stroke (ICC ≥ 0.86–0.95) [25] and older adults (ICC ≥ 0.98) [42].

The five-times sit-to-stand test (5STS) [35] was used in this study, to evaluate the strength of lower limbs. The 5 s chair stand (5STS) test was conducted using a standard chair height. Participants were explicitly told not to utilize their arms during the test. They were instructed to “rise from the chair and then promptly return to a seated position for five repetitions, refraining from using their hands”. The evaluator ensured that participants stood up fully (achieving complete extension) and sat down (making contact with the chair) for each repetition [35]. A single trial of the 5STS test was executed, and the time taken to accomplish the five repetitions was recorded using a stopwatch.

The 5STS measures the time taken, in seconds, to complete five repeated chair stands. For older adults and patients with chronic stroke, the cutoff score is 12 s [35]. 5STS is a clinical tool that is simple to administer with an excellent intrarater (ICC = 0.97), interrater (ICC = 0.99), and test–retest reliability (ICC = 0.994) [43].

#### 2.3.3. Muscle Mass: Calf Circumference

Calf circumference (CC) is acknowledged as a measure of muscle mass in older adults by the World Health Organization [36]. It is an easy and quick screening method, based on the correlation between CC and human muscle mass [26]. A flexible tape ruler with an accuracy of 0.1 cm was used to measure the CC. The participant sat on the chair with the knee bent to 90 degrees and the sole of the foot resting on the floor. The flexible tape was positioned perpendicular to the leg axis and then wound around it. Subsequently, the highest circumferential measurement (CC) was documented [44]. Both calves were measured, in the thickest part of the lower legs holding the flexible measuring tape in contact with the skin but taking care not to compress the soft tissue. A cut of ≤31 cm has been shown to predict performance and survival in older people [8]. The variation in CC between right and left sides was calculated. Moreover, several studies have indicated that the CC measurement has been effective in screening for sarcopenia among older adults living in the community and older stroke patients [44]. These findings suggest that CC could potentially serve as a valuable tool for identifying cases of sarcopenia.

#### 2.3.4. Gait Speed: 10 m Walk Test (10 MWT)

The 10 m walk test (10 MWT) has been widely utilized to assess the walking capacity of people with stroke [29]. During the test, the participant was instructed to walk at a comfortable speed along a 10 m walkaway. Timing was performed during the central 6 m, allowing 2 m at the beginning and 2 m at the end for acceleration and deceleration, respectively. The patient receives guidance to walk at their own chosen pace, incorporating any required walking aids like a walker or cane. The velocity is determined by dividing the distance covered by the time taken [29]. Gait speed has been demonstrated that it can reliably predict an individual’s community walking status [45]. Specifically, a gait speed of less than 0.4 m/s predicts walking primarily within the household, a speed between 0.4 and 0.8 m/s predicts limited community walking, and a speed exceeding 0.8 m/s predicts unlimited community walking. The test has shown excellent reliability ICC 0.94 [46].

#### 2.3.5. Functionality: Short Physical Performance Battery

The SPPB [37] evaluates three distinct functional components: balance, gait, and leg strength. The standing balance component measures the ability to maintain balance for 10 s in three different foot positions: side by side, semi-tandem, and tandem. The gait speed component assesses gait speed during a 4 m walk test. The leg strength component measures the time taken to complete five sit-to-stand repetitions. Each component is scored on a scale of 0 to 4 points, with a maximum total score of 12. The remarkable sensitivity of the SPPB, demonstrated by its cut-point of ≤8 points, indicates its potential as a valuable screening tool for sarcopenia in clinical settings [47]. The SPPB has shown high reliability (ICC = 0.92) [37].

### 2.4. Simple Size Estimation

An a priori sample size calculation was conducted in G* Power software (version 3.1.9.4; Heinrich-Heine-Universität, Düsseldorf, Germany). For a two-tailed *t*-test, assuming a medium/large effect size d = 0.70, α = 0.05 and β = 0.20, a minimum sample size of 34 participants per group was required. A similar sample size for measuring post-stroke sarcopenia variables has been previously used for this population [48].

### 2.5. Statistical Analysis

All statistical analyses were performed in SPSS version 26.0 (IBM, Armonk, NY, USA). Descriptive results are presented as mean (standard deviation), median (25–75th percentile) or frequencies (percentage), as appropriate. Shapiro–Wilk tests were used to check the normality of the data.

To analyze the between groups differences, the Mann–Whitney U test, independent *t*-test, or the chi-square (χ^2^) test were used. The effect size was interpreted as small (d = 0.2), medium (d = 0.5), and large (d > 0.8) [49].

## 3. Results

A total of 68 participants were included and recruited in this study (SG, n = 34 and CG, n = 34). Table 1, presents demographics, anthropometrics, and clinical characteristics by group. The participants with chronic stroke showed a median modified Rankin scale of 2 (slight disability) and a significantly higher comorbidity index and lower cognitive function in comparison to the CG.

### 3.1. Sarcopenia-Related Variables in the Chronic Stroke Group Compared to Non-Stroke Counterparts

The between-group comparison results for sarcopenia screening (SARC-F), muscle strength (hand grip and 5STS), CC, and functionality (SPPB) are shown in Table 2. Maximum hand grip strength was the only variable that did not show statistically significant differences when both groups were compared. For the SG, the screening test for sarcopenia showed a median of 3 points, thus, near the cut-off which indicates risk of sarcopenia (SARC-F ≥ 4). Moreover, for the SG the median values of the 5STS (16.26 s) and SPPB (7 points) were over and below, respectively, to the cutoff values. In all cases there were significant differences between groups and the effect size was medium or large. The CG obtained a scores over the cutoff points for each of the variables related to sarcopenia.

### 3.2. Sarcopenia-Related Variables of Stroke and Non-Stroke Participants Depending on Their Age Group

When participants with stroke and without stroke were compared by age the results revealed that there was a statistically significant difference in all the variables studied except maximum hand grip strength, and except for the CC in the group over 65 years of age (Table 3).

In relation to the screening test for sarcopenia (SARC-F), 34.8% of participants with chronic stroke aged between 40–65 presented a high probability of sarcopenia (scores ≥ 4) and in the case of the group over 65 was 45.5%. In the CG, the participants of both age groups showed no risk of suffering from sarcopenia.

It can be observed that the group > 65 years with chronic stroke compared to older adults without stroke showed values for 5STS, SPPB, and 10 MWT within those considered as having risk of sarcopenia or close to them. In the case of the participants between 40 and 65 years old with stroke, it was observed that both the 5STS and the 10 MWT showed values that were over the cut-off, so they had risk of sarcopenia. The differences between groups for these variables were statistically significative.

Regarding maximum hand grip strength, there are no statistically significant differences when comparing the SG and CG by age group. However, when comparing the strength difference between both upper limbs, we found statistical differences both in the 40–65 age group and in the over 65 group. In the descriptive analysis of hand grip strength according to sex and age (Figure 1), it was observed that older women over 65 years old, both in the SG and CG, did not reach the minimum score for this variable (hand grip strength <16 kg). As for men, the group of participants with stroke over 65 years old scored below the cut-off point (hand grip strength <27 kg).

## 4. Discussion

The results of this study show that the measurement of sarcopenia variables has different characteristics in chronic stroke survivors compared to non-stroke counterparts, both for the 40–65 years old group and for the over 65 years old group, with significant differences in their assessment scores. Moreover, the tools for assessing sarcopenia have a different impact in their screenings for stroke and non-stroke participants. The five-times sit-to-stand test, SPPB, and gait speed can lead clinicians to detect stroke-related sarcopenia. Maximum handgrip both in women and men over 65 years old shows a trend of low values for the SG. However, CC did not detect sarcopenia in our sample.

Following the steps in the assessment to detect sarcopenia, the initial screening tool is the SARC-F. In the current study, the final scores were significantly different between SG and CG, with the SG having a higher score which translates into worse performance in the items of the test. This can be explained by the fact that items in the SARC-F include five self-reported questions answered by the patients themselves in relation to strength, walking, rising from a chair, climbing stairs, and falls, and all of these outcomes are motor tasks which can be highly disturbed in people with chronic stroke [50]. However, in the median values, both groups (SG and CG) had scores under the cut-off, which means, they had low risk of sarcopenia. In line with this results, previous research has shown that when applying the SARC-F in older adults, case finding of sarcopenia decreases [51], so using the SARC-F can be at the expense of missing cases. Moreover, this questionnaire has shown very good specificity to diagnose sarcopenia, but a low sensitivity [30,52]. The EWGSOP2 recommends case finding should start when a patient reports symptoms or signs of sarcopenia, such as falling, difficulty in rising from a chair, or slow walking speed [53], which are all aspects that are highly affected in chronic stroke [54]. In fact, in our sample, the SG showed a low walking speed with differences with the CG. Therefore, in cases [53] where clinical signs are observed, there is no need to use any screening questionnaire, and further tools for testing for sarcopenia can be used, and this could also apply in chronic stroke rehabilitation units in light of our results.

In regard to muscle strength, hand grip and the five-times sit-to-stand test were used to assess upper and lower limb strength, respectively. There were no statistical differences between the SG and CG in the maximum handgrip, although globally the scores were lower in the SG, which may indicate that there is a trend of low strength in the SG. This is clinically useful since previously it has been stated that hand grip correlates well with overall strength and function of the upper limb in patients with stroke [25] in the acute stage, therefore, it reinforces the idea of including this assessment also in the chronic stage. When limb differences were analyzed in regard to hand grip, it showed that there were also statistically significant differences between both groups. Considering that hand grip is related to force production [55], which is needed for motor rehabilitation of the upper limb of people with stroke, the fact that the SG had higher differences between upper limbs has clinical implications. It could imply differences on the use of the paretic and non-paretic upper limb in daily activities and, therefore, influence the long-term capacity of chronic stroke patients [56]. In the clinical setting, this is suggestive of the importance of evaluating hand grip in both limbs even in the chronic stage, since differences are notable and should be addressed in the treatment.

In the five-times sit-to-stand test, there were significant differences in the median values between SG and CG, finding this was below the cut-off point in the SG, indicating low strength. In terms of sarcopenia, this means it is detecting probable sarcopenia, and, in clinical practice, this is enough to start intervention [8] since it is better to prevent loss of strength and function rather than trying to restore it when it has progressed [57]. Moreover, in rehabilitation the ability to stand up from a chair is an essential prerequisite for mobility and functional independence [58], so it is an essential assessment, plus it can influence further motor tasks in the maintenance process.

Regarding muscle mass, the CC was used for assessment. Although the EWGSOP2 recommends the use of other techniques, in routine clinical examinations these techniques have a high cost and a radiation exposure [59]. Moreover, the CC measurements can be used as a substitute for diagnosis for older adults in settings where there are no other methods of muscle mass diagnosis available [8]. In our results, there were statistical differences between the SG and the CG, with the SG having a higher average score. This could be explained by the fact that the SG had a higher BMI, in fact, it was statistically different to the CG. Moreover, the values of the SG in relation to their mean BMI indicate they are overweight, near to the obesity value. Therefore, in the SG, there might have been an increase in the intramuscular fat [60] which could explain the values of the CC. Moreover, when the limb differences of the CC are analyzed, there are changes between SG and CG. In healthy individuals, the variation in CC between right and left sides is typically less than 1 cm in 90% of cases and less than 1.5 cm in 100% of cases [61] which concur with the results of the CG. However, a significant asymmetry of the CC could indicate atrophy of the smaller sides or edema of the other side [61]. Therefore, future studies could analyze the differences in lower limb in the CC assessments to deepen our results.

As for functionality, both the SPPB and 10 m walk test showed differences statistically between the SG and the CG, with the SG showing scores below the cut-off. In terms of sarcopenia, the EWGSOP2 indicates that a low physical performance confirms severe sarcopenia. This was as expected, since the prevalence of sarcopenia after a stroke is associated with poor functional outcomes [62], and when stroke-related sarcopenia [6] is identified it has also been associated with worse clinical outcomes and physical dysfunction.

In relation to the differences between age groups, for the SARC-F, there were significant differences in both age groups between SG and CG, with the SG having higher scores, and the older adults being near the cut-off. However, again, none of the median scores were over the cut-off, although nearly half of the SG older adults showed risk of sarcopenia with the SARC-F. Therefore, the SARC-F questionnaire can be a means to obtain self-reports from patients in clinical practice, but clinicians do not need to feel obliged to use SARC-F [53], rather any symptom related to sarcopenia should prompt the clinician to assess for it, even more in chronic stroke.

Regarding maximum hand grip in relation to age groups the assessment scores of the SG showed that women and men over 65 years old had low strength (below the cut-off). Stroke has been, in fact, associated with advanced age so it has also brought recent attention to the potential impact of aging on hemiparetic muscle [7]. Women in the CG over 65 years old also had hand grip below the cut-off, but not men. Our results highlight the importance of age, but there might also be sex differences that may have to be brought into attention in the treatment of chronic stroke. However, larger samples are required to corroborate this trend.

In relation to the five-times sit-to-stand test, there were statistical differences in both age groups, with the SG having scores over the cut-off, regardless of the age. Therefore, in the SG participants of both age groups had probable sarcopenia according to the five-times sit-to-stand test, which reinforces the idea that this is a stroke-related sarcopenia. Hemiplegia resulting from stroke pyramidal tract disorder leads to a combination of disuse, denervation, remodeling, inflammation, and spasticity leading to a complex pattern of change of the muscle phenotype and its subsequent atrophy [7], which explain the observed stroke-related sarcopenia.

In relation to CC compared by age groups, there were statistical differences between the measurement in favor of the SG in both age groups. The CC has shown to be accurate for screening stroke-related sarcopenia [26], however, it was based on the Asian Working Group for Sarcopenia 2019 diagnostic criteria. In a previous study conducted of the thigh muscles, it was stated that the muscle area and muscle volume of the hemiplegic thigh of stroke survivors were 20–24% lower than that of nonparetic thighs, and the intramuscular fat was 17–25% higher than that of nonparetic thighs [63]. Moreover, the increase in intramuscular fat can significantly affect the volume of lean tissue [64]. Therefore, as stated before, there is a need of further research to elucidate its clinical impact.

In physical performance, both the SPPB and gait speed showed differences between the SG and the CG according to age groups. When participants had stroke, regardless of the age, the functionality was affected, thus, having severe sarcopenia. SPPB has emerged as a promising tool to evaluate functional capability [65] and gait speed is a quick and easy method to evaluate physical performance in both sarcopenia and stroke patients [66]. Therefore, in light of our results, in clinical settings in order to assess stroke-related sarcopenia these are both tools which may help clinicians.

### Limitations and Future Directions

As for limitations of this study, the first is not having used an instrumented diagnostic measure for muscle quantity and quality. However, our aim was precisely to use patient-friendly instruments. Another limitation is that the sample was recruited only in one region, therefore, larger multicentric samples would be recommended. Moreover, although the sample size was adequate, a larger sample size would be recommended in future studies due to the heterogeneity related to stroke and could allow studying sex and age subgroups, initial NIHSS of stroke, time of evolution since stroke occurred, or applied treatments.

Overall, although the body disability after stroke leads to difficulties in the screening and diagnosis of sarcopenia, this study has identified the impact the different tools may have in the clinical assessment of stroke-related sarcopenia. The complex metabolic processes after stroke, which contribute to tissue wasting and development of sarcopenia, and their impact on functional capacity, are incompletely understood, so further studies in stroke survivors are needed. For future studies it would be interesting to propose appropriate physical and nutritional strategies in stroke rehabilitation in the chronic stage of stroke.

## 5. Conclusions

Considering the muscle atrophy and weakness due to the pathology process in chronic stroke, it is clinically important to identify assessment tools for detecting stroke-related sarcopenia. In relation to the impact of the tools for assessing sarcopenia, our results have shown that the five-times sit-to-stand test, the SPPB, and the 10 MWT can lead clinicians to detect stroke-related sarcopenia and its severity. Maximum handgrip shows a trend of low values for the SG, which could help researchers establish new study lines related to chronic stroke. However, CC did not detect sarcopenia in our sample. Globally, the measurement of sarcopenia variables shows different characteristics in chronic stroke survivors compared to non-stroke counterparts, both for the 40–65 years old group and for the over 65 years old group, with significant differences in their assessment scores. In chronic stroke survivors, assessing variables related to sarcopenia has to be considered in order to propose appropriate strategies in the chronic stage of stroke rehabilitation.

## Figures and Tables

**Figure 1 biomedicines-11-02601-f001:**
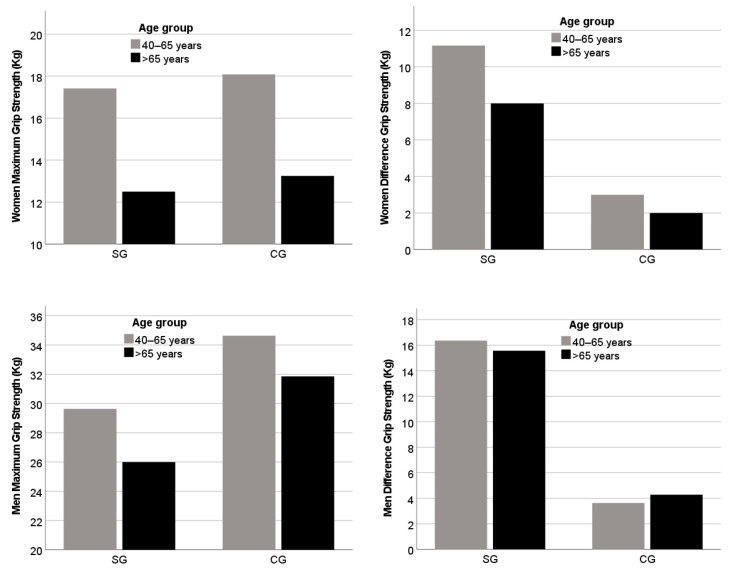
Description of maximum hand grip strength and limb difference in hand grip strength by sex and group.

**Table 1 biomedicines-11-02601-t001:** Clinical characteristics of participants by group.

	Stroke Group (n = 34)	Control Group (n = 34)	Between-Group Differences (*p*-Value; Effect Size)
Demographics and anthropometrics			
Age	60.74 (9.73)	60.85 (9.72)	0.96; -
Women/men, n (%)	16 (47.1)/18 (52.9)	16 (47.1)/18 (52.9)	1.00; -
BMI	28.72 (4.73)	25.31 (3.65)	**0.001**; 0.81
Clinical characteristics			
Comorbidity—CCI score	3.0 (1.0–6.0)	0.0 (0.0–2.0)	**<0.001**; 2.91
Cognitive status—MOCA	21.74 (4.53)	26.35 (2.57)	**<0.001**; 1.25
Specific characteristics for stroke			
Time since stroke (months)	55.0 (9.0–224.0)	-	
Type of stroke: Ischemic/hemorrhagic, n (%)	22 (64.7)/12 (35.3)	-	
Side of hemiparesis: left/right, n (%)	13 (38.2)/21 (61.8)	-	

Data are expressed as mean (SD), median (min–max), or otherwise stated. Significant differences are highlighted in bold. Between-group differences were calculated with independent *t*-test (t) or the Mann—Whitney (U) test for continuous data, and the chi-squared test (χ^2^) for categorical data. BMI: body mass index; CCI: modified Charlson comorbidity index; MOCA: Montreal cognitive assessment.

**Table 2 biomedicines-11-02601-t002:** Sarcopenia-related variables compared by group.

	Stroke Group (n = 34)	Control Group (n = 34)	Between-Group Differences (*p*-Value; Effect Size)
Sarcopenia Screening			
SARC-F	3.00 (0.00–8.00)	0.00 (0.00–2.00)	**<0.001**; 1.83
Muscle Strength			
Maximum hand grip strength (kg)	22.56 (9.82)	25.71 (10.17)	0.20; -
Women	18.00 (2.00–22.00)	18.00 (8.00–24.00)	0.87; -
Men	28.22 (9.35)	33.56 (6.30)	0.05; -
Limb difference hand grip strength (kg)	12.00 (0.00–38.00)	2.50 (0.00–12.00)	**<0.001**; 1.30
Women	9.50 (1.00–20.00)	2.00 (0.00–9.00)	**0.003**; 1.21
Men	16.06 (12.35)	3.89 (3.55)	**0.001**; 1.34
Five-times sit-to-stand (sec)	16.26 (7.88–38.26)	11.27 (6.42–20.63)	**<0.001**; 1.38
Muscle Mass			
Maximum calf circumference	38.76 (3.22)	36.74 (3.67)	**0.018**; 0.59
Difference between limbs calf circumference	1.65 (0.00–5.50)	0.75 (0.00–3.00)	**0.001**; 0.82
Functionality			
SPPB (0–12 score)	7.00 (1.00–12.00)	11.00 (9.00–12.00)	**<0.001**; 1.45
10 MWTAuto (m/s)	0.72 (0.38)	1.40 (0.16)	**<0.001**; 2.33

Data are expressed as mean (SD) or median (min–max). Significant differences are highlighted in bold. Between-group differences were calculated with independent *t*-test (t) or the Mann—Whitney (U) test. SARC-F: strength, assistance in walking, rise from a chair, climb stairs, falls history questionnaire; SPPB: short physical performance battery; 10 MWT: 10 m walk test, auto select speed.

**Table 3 biomedicines-11-02601-t003:** Sarcopenia-related variables compared by group and age.

	Group 40–65 Years			Group > 65 Years		
	Stroke Group (n = 23)	Control Group (n = 23)	Between-Group Differences (*p*-Value; Effect Size)	Stroke Group (n = 11)	Control Group (n = 11)	Between-Group Differences (*p*-Value; Effect Size)
Sarcopenia Screening						
SARC-F	2.00 (0.00–8.00)	0.00 (0.00–2.00)	<0.001; 1.77	3.00 (0.00–8.00)	0.00 (0.00–2.00)	**<0.001**; 2.17
Muscle Strength						
Maximum hand grip strength (kg)	21.00 (7.00–47.00)	24.00 (9.00–42.00)	0.29; -	21.09 (11.61)	25.09 (11.50)	0.21; -
Limb difference hand grip strength (kg)	13.00 (1.00–36.00)	3.00 (0.00–10.00)	**<0.001**; **1.44**	10.00 (0.00–38.00)	2.00 (0.00–12.00)	**0.047**; 0.91
Five-times sit-to-stand (sec)	16.26 (7.94–28.97)	11.34 (6.42–20.63)	**<0.001**; 1.46	16.93 (7.88–38.26)	11.08 (8.60–15.08)	**0.016**; 1.19
Muscle Mass						
Maximum calf circumference	39.17 (2.96)	36.76 (4.11)	**0.027**; 0.67	37.91 (3.72)	36.68 (2.71)	0.39; -
Difference between limbs calf circumference	1.00 (0.00–5.00)	1.00 (0.00–1.50)	**0.026**; 0.67	2.00 (0.00–5.50)	0.00 (0.00–3.00)	**0.034**; 1.00
Functionality						
SPPB (0–12 score)	8.00 (1.00–12.00)	11.00 (9.00–12.00)	**<0.001**; 2.08	5.00 (2.00–12.00)	11.00 (9.00–12.00)	**0.003**; 1.54
10 MWT Auto (m/s)	0.76 (0.37)	1.42 (0.13)	**<0.001**; 2.38	0.63 (0.39)	1.35 (0.20)	**<0.001**; 2.32

Data are expressed as mean (SD) or median (min–max). Significant differences are highlighted in bold. Between-group differences were calculated with independent *t*-test (t) or the Mann—Whitney (U) test. SARC-F: strength, assistance in walking, rise from a chair, climb stairs, falls history questionnaire, SPPB: short physical performance battery; 10 MWT: 10 m walk test, auto select speed.

## Data Availability

The data are available under reasonable request.

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
