# Peer review of "Assessing Stroke-Related Sarcopenia in Chronic Stroke: Identification of Clinical Assessment Tools—A Pilot Study"

_biomedicines, 2023, doi:10.3390/biomedicines11102601_

Round 1

Reviewer 1 Report

Arnal-Gomez and colleagues, in the present research article titled ‘Identifying tools for clinical assessments of stroke-related sar-2 copenia in chronic stroke: a cross-sectional study’ investigate the assessment of sarcopenia in chronic stroke survivors and compares it to non-stroke counterparts. Sarcopenia, the loss of muscle strength and mass with age, is particularly prevalent in stroke survivors and contributes to chronic disability. However, there is limited clinical literature on post-stroke sarcopenia in the chronic stage. The study involved a cross-sectional analysis of a total of sixty-eight participants, divided into a chronic stroke group (SG) and a non-stroke control group (CG). Various tools were used to assess sarcopenia variables, including the SARC-F questionnaire for screening, hand grip strength, five times sit-to-stand test, calf circumference measurement, gait speed, and the Short Physical Performance Battery (SPPB). The results indicate significant differences in the measurement of sarcopenia variables between the chronic stroke group and non-stroke counterparts, indicating the impact of stroke on muscle function and mass. The Five Sit to Stand test, SPPB, and gait speed were found to be effective tools in identifying stroke-related sarcopenia. However, the calf circumference measurement did not show consistent results in detecting sarcopenia. The study highlights the importance of addressing sarcopenia in chronic stroke rehabilitation and suggests the need for appropriate physical and nutritional strategies to improve functional capacity and quality of life for stroke survivors. The research identifies an area of research gap, providing valuable insights into the assessment and management of sarcopenia in chronic stroke patients.

In general, I think the idea of this article is really interesting and the authors’ fascinating observations on this timely topic may be of interest to the readers of Biomedicines. However, some comments, as well as some crucial evidence that should be included to support the author’s argumentation, needed to be addressed to improve the quality of the manuscript, its adequacy, and its readability prior to the publication in the present form. My overall judgment is to publish this paper after the authors have carefully considered my suggestions below, in particular reshaping parts of the ‘Introduction’ and ‘Methods’ sections by adding more evidence.

 Please consider the following comments:

I recommend revising the title, as it could be more concise to make it easier for readers to understand at a glance. Consider shortening the title while maintaining its essential components. A potential revised title could be: "Assessing Stroke-Related Sarcopenia in Chronic Stroke: Identification of Clinical Assessment Tools" [1-3].

A graphical abstract that will visually summarize the main findings of the manuscript is highly recommended.

Abstract: According to the Journal’s guidelines, this section should be presented as a short summary of about 200 words maximum that objectively represents the article. It should let readers get the gist or essence of the manuscript quickly, prepare the readers to follow the detailed information, analyses, and arguments in the full paper and, most of all, it should help readers remember key points from your paper. Please, consider rewrite this paragraph following these instructions [4]. 

Keywords: Please list ten keywords chosen from Medical Subject Headings (MeSH) and use as many as possible in the title and in the first two sentences of the abstract. I would suggest adding “Motor impairment” and “Physical disability” as keywords.

Introduction: The authors need to reorganize this section with several paragraphs made up of about 1000 words, introducing information on the main constructs of this study, which should be understood by a reader in any discipline, and making it persuasive enough to put forward the main purpose of the current research the authors have conducted and the specific purpose the authors have intended by this protocol. I would like to encourage the authors to present the introduction starting with the general background, proceeding to the specific background on understanding of stroke-related sarcopenia, its prevalence, and the need for further research on its assessment and impact in the chronic stage of stroke. Those main structures should be organized in a logical and cohesive manner [5]. 

In this regard, I believe that the Introduction section would benefit from additional information to enhance its clarity and contextualization. To strengthen this section, I believe that it is essential to consider the neural substrates that contribute to the impairment of motor function and muscle atrophy in stroke survivors. Stroke often results in brain injury, particularly affecting areas associated with motor control and movement. The neural substrates play a crucial role in the recovery of motor function and the development of sarcopenia in stroke patients. One of the key neural substrates affected in stroke is the descending drive, which refers to the neural signals sent from the brain to the spinal cord and peripheral muscles to initiate and control movement. Stroke-induced brain damage can disrupt these neural pathways, leading to impaired motor control and reduced muscle activation. Moreover, disuse and compensatory motor patterns may arise due to decreased physical activity following stroke, contributing to muscle atrophy and weakness [6-7]. These changes can lead to a loss of muscle mass and strength, a phenomenon referred to as "stroke-related sarcopenia." In addition to the impairment of descending drive and disuse, spasticity is another crucial neural substrate that plays a role in the development of sarcopenia in stroke patients. Spasticity refers to the involuntary contraction of muscles, leading to increased muscle tone and resistance to passive movement. The continuous spastic muscle activity can cause muscle fatigue and atrophy over time, further exacerbating the muscle weakness seen in stroke-related sarcopenia. Understanding the neural substrates involved in the pathophysiology of stroke-related sarcopenia is crucial for designing effective rehabilitation strategies and interventions. Addressing these neural mechanisms can aid in optimizing motor recovery and minimizing the adverse effects of sarcopenia in chronic stroke survivors [8-10]. In summary, including a discussion of neural substrates in the introduction will provide a comprehensive perspective on the complexities of stroke-related sarcopenia and emphasize the importance of considering both neural and muscular aspects in the evaluation and management of this condition.

Material and Methods: I believe that this section would benefit from a clearer structure and better organization of the flow of information. For example, while the section briefly describes the measurement tools used for sarcopenia assessment (SARC-F, hand grip, five times sit to stand test, calf circumference, gait speed, and Short Physical Performance Battery), additional information is needed. It's crucial to provide references for the specific protocols and guidelines used for each measurement tool and discuss their validity and reliability in the context of stroke-related sarcopenia. Furthermore, here the Authors mention that an a priori sample size calculation was performed, but it would be helpful to elaborate on the statistical assumptions used and how the sample size of 68 participants was determined to be sufficient for the study's objectives.

In my opinion, the ‘Conclusions’ paragraph would benefit from some thoughtful as well as in-depth considerations by the authors, because as it stands, it lists down all the main findings of the research, without really stressing the theoretical significance of the study. Authors should make an effort, trying to explain the theoretical implication as well as the translational application of their research.

In according to the previous comment, I would ask the authors to include a proper and defined ‘Limitations and future directions’ section before the end of the manuscript, in which authors can describe in detail and report all the technical issues brought to the surface,

I hope that, after these careful revisions, the manuscript can meet the Journal’s high standards for publication. I am available for a new round of revision of this article. 

Best regards,

Reviewer

References: 

1. https://plos.org/resource/how-to-write-a-great-title/

2. https://www.nature.com/nature-index/news-blog/how-to-write-a-good-research-science-academic-paper-title

3. https://www.indeed.com/career-advice/career-development/catchy-title

4. https://www.mdpi.com/journal/biomedicines/instructions

5. https://dept.writing.wisc.edu/wac/writing-an-introduction-for-a-scientific-paper/

6. DOI: 10.17219/acem/165944 

7. https://doi.org/10.3390/ijms24065926

8. DOI: 10.3390/biomedicines11030945

9. https://doi.org/10.3389/fnmol.2023.1217090

10. https://doi.org/10.3390/biomedicines11051248

Minor editing of English language required.

Author Response

In relation to the manuscript with title ‘Identifying tools for clinical assessments of stroke-related sarcopenia in chronic stroke: a cross-sectional study’ we would like to thank the reviewer for the comments on our manuscript. For sure the aspects that have been changed due to the comments will help improve the understanding of our research and its impact.

Reviewer 2 Report

This is an interesting study, but there are some problems.

What is the hypothesis of this study at the end of the introduction?

How many did the sample size need used muscle mass?

Please provide the STROBE form.

The initial NIHSS of stroke is important for muscle power, and the duration and method of physical therapy also influence the improvement of muscle power.  The study included older patients who suffered from more than 2 years after stroke, it may be the age effect that needs to include no stroke older to adjust the risk using the odds ratio. 

Author Response

(The authors gave the same response as above.)

Round 2

Reviewer 1 Report

Dear Authors,

I am pleased to acknowledge that you have indeed addressed all of my concerns and queries in a clear and precise manner. Your responses have provided valuable insights into the modifications made to the manuscript in light of my comments. It is evident that you have taken great care to ensure that the revised manuscript aligns more closely with the scientific rigor expected for publication in Biomedicines.

Upon reviewing the updated version, I find that the inclusion of the additional studies has indeed enriched the understanding of neural substrates that contribute to the impairment of motor function and muscle atrophy in stroke survivors. The provided studies contribute significantly to the comprehensiveness of the section. However, in order to provide a more holistic view of the complex phenomena underlying the recovery of motor function and the development of sarcopenia in stroke patients, I believe there's still an opportunity to expand upon certain factors. Specifically, the discussion of the pathophysiology of stroke-related sarcopenia would offer a deeper insight into the mechanisms at play (DOI: 10.3390/biomedicines11030945; https://doi.org/10.3389/fnmol.2023.1217090; DOI: 10.3390/cells11162607). By incorporating these aspects, the Introduction section would offer a comprehensive overview of the complexities of stroke-related sarcopenia and emphasize the importance of considering both neural and muscular aspects in the evaluation and management of this condition.

I want to reiterate my appreciation for your responsiveness and willingness to consider these suggestions. I believe that this minor revision will significantly enhance the quality and impact of the Introduction section. 

Thank you once again for your dedication to improving the manuscript. I look forward to seeing the continued progress.

Author Response

Thank you for your comments. 

Reviewer 2 Report

The muscle atrophy may be caused by stroke, but authors did not provide initial muscle power and NIHSS.

The selection bias was severe by author selection, the random control trial must collect.

This study was not benefit for stroke care at all. It was not quality to publish,

Author Response

Thank you for your comments. 
